# Biophysical Benefits Simulation Modeling Framework for Investments in Nature-Based Solutions in São Paulo, Brazil Water Supply System

Eileen Andrea Acosta [1,2,]* , Se Jong Cho [3,4] , Claudio Klemz [1] , Justus Reapple [1] , Samuel Barreto [1] , Bruna Stein Ciasca [1] , Jorge León [5] , Carlos Andres Rogéliz-Prada [1] and Henrique Bracale [1]

1   The Nature Conservancy, Global Office, Arlington, VA 22203, USA
2   Water Resources and Environmental Engineering, Federal University of Paraná, Curitiba 81530-000, PR, Brazil
3   The National Socio-Environmental Synthesis Center, University of Maryland, Annapolis, MD 21401, USA
4   US Geological Survey, Water Resources Mission Area, Reston, VA 20192, USA
5   Center for Nuclear Energy in Agriculture, University of São Paulo, Piracicaba 13416-000, SP, Brazil
*   Correspondence: eacosta@tnc.org

**Abstract:** In order to understand the hydrological impacts of the nature-based solutions in the Cantareira Water Supply System, this study evaluates six different land cover and land use change scenarios. The first and second consider the restoration of native vegetation in riparian areas, the third prioritizes restoration sites using biophysical characteristics (optimized restoration scenario derived from Resource Investment Optimization System—RIOS), the fourth considers best management practices and the fifth and sixth are hypothetical extreme scenarios converting all pasture to forest and vice versa. Two hydrological models were developed to represent the distributions of water and yields in the study watershed: HEC-HMS and SWAT. Simulation results indicate that when nature-based solutions are implemented, surface runoff is reduced and ambient storage increases during the rainy season (December–March); while the overall flow increases during the dry season (June–September). The combination of specific hydrologic components of RIOS-customized intervention scenario simulation outputs—namely surface flows and groundwater contribution to stream flows—indicate on average 33% increase in the overall water yield, or 206 hm$^3$/year, across the study watershed when comparing against the baseline conditions. In the same modeling scenario, the water storage in the sub-watersheds adjacent to the reservoirs showed an increase of 58% (or 341 hm$^3$/year). The results indicate that adopting NbS in the source watershed can mitigate the impacts of extreme drought conditions and contribute toward building long-term water security.

**Keywords:** nature-based solutions; water security; hydrologic model; SWAT; HEC-HMS; landscape scenarios

## 1. Introduction

More than half of the world's population lives in cities, and their source watersheds are often degraded due to poor landscape management and conversion of natural environments for urban sprawl. Population concentration, inefficient use of water, economic development, as well as climate change influences, can cause a water security risk. According to the World Economic Forum (WEF) and UN's Sustainable Development Goal #6, securing an adequate supply of clean water despite the damaging effects of climate change is one of the world's most urgent challenges. At the WEF global risk reports, extreme weather water and climate action failure are ranked among the top 10 risks for the global economy [1]. In fact, for the past 10 years, WEF Global Risks reports have noted water supply crises and water hazards related to extreme weather events as the most significant global risks in terms of likelihood and severity of impact, with biodiversity loss and ecosystem degradation likewise consistently ranking in the top risk categories [2].

As climatic extremes become more frequent with greater magnitudes, severe droughts have been observed in the dry season and floods and landslides in the rainy season. For instance, in the Metropolitan Region of Sao Paulo (RMSP), a repeated combination of drought and flood conditions threaten the water supply system, the ecosystem, the economy, and its population. The region experienced the worst drought in 80 years in the last decade and has continued to face water supply crisis [3]. Specifically, the Cantareira Water Supply System (CWSS)—which consists of 2200 km$^2$ of source watersheds with four interconnected reservoirs (Figure 1)—was one of the most affected in the 2014–2015 drought. The CWSS is the largest producer of water in the RMSP, with storage capacity of about 1000 hm$^3$, supplying water to approximately 46% of the population (nearly 9 million people).

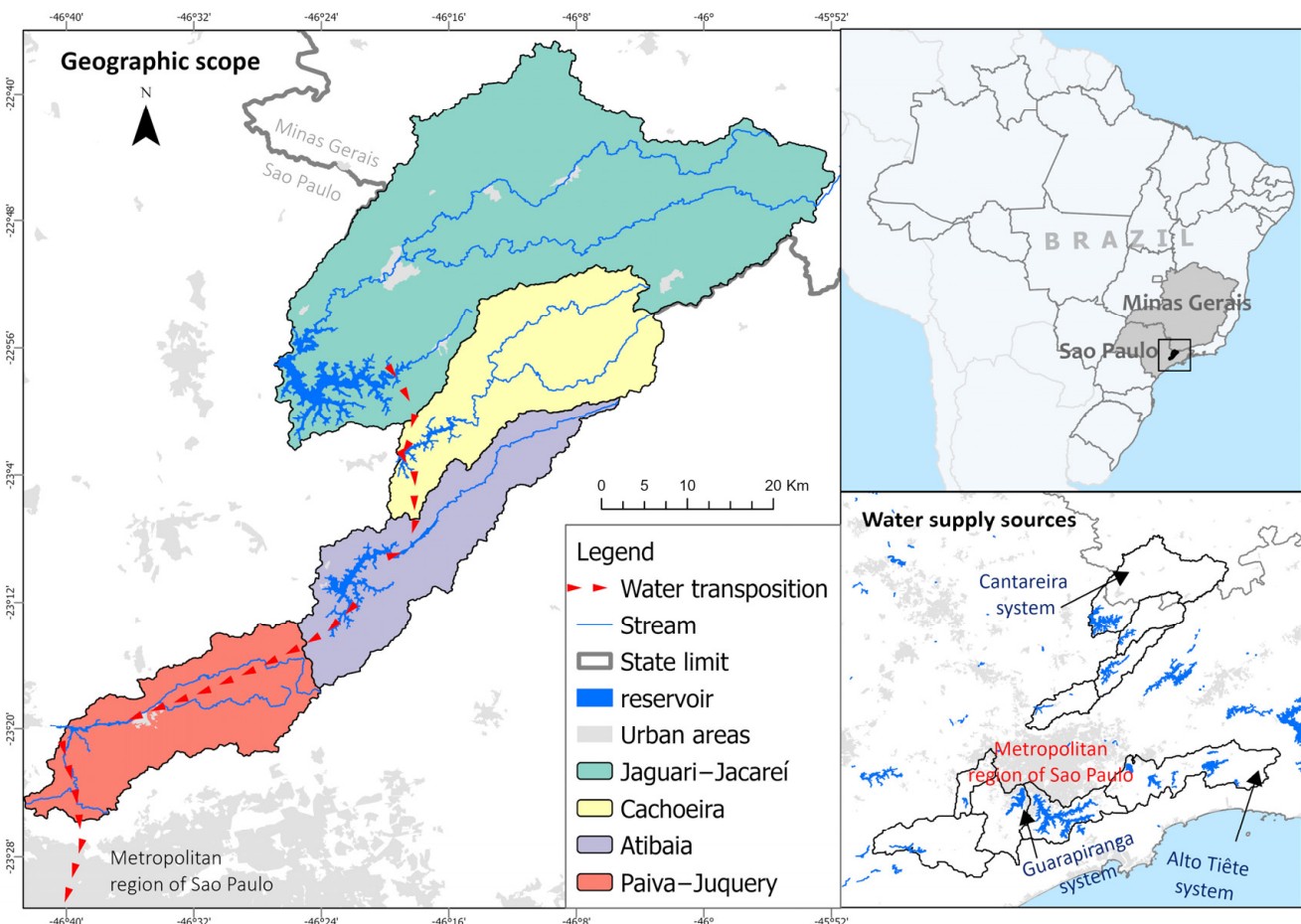

**Figure 1.** Geographic scope includes the sub-watersheds and reservoirs of the Cantareira system. Red dotted line shows the direction of water transposition. Map frame shows the localization and the main water sources for the Metropolitan region of São Paulo.

Conventional water resource management strategy has focused mainly on engineered interventions (e.g., grey infrastructures such as reservoirs, check dams, inter-watershed transfers, etc.), which are limited in its capacity to guarantee water security with the changing climate and growing water demands across the world [4]. The viability of new water supply systems also diminishes as conflicts emerge with other large cities and regions that also face growing water demands. Therefore, conservation and restoration of source watersheds are essential to building more resilient water supply system, in which nature-based solutions (NbS) emerge as an important option in policy objective for water security. To mainstream NbS application in source watersheds, decision makers need tools for integrating science-based knowledge [5] that can build shared understanding of the intrinsic landscape characteristics and its hydrology, and develop mutual language to establish water resource management goals.

Experimental studies with monitoring data investigating the effects of NbS implementation in the watershed demonstrate that landuse/landcover (LULC) changes affect hydrologic balance and water yield. There is also an evidence of decreases in peak discharge at the catchment outflow [6], increases in water storage in the soil columns [7] and the improvements in water quality [8]. Data from remote sensing from 2000 to 2015 shows an increase in permanent surface water in vegetation restoration regions [9].

Various studies have shown that NbS can reduce surface runoff, favoring infiltration and decreasing soil erosion, contributing to establishing greater climatic resilience in water resources management [10,11]. Thus, to better (i) assess optimal allocation of NbS interventions, (ii) predict changes in surficial and groundwater water balance, and (iii) simulate the long-term water availability and storage in the CWSS, we applied a set of biophysical models to simulate the effects of landscape restoration on multiple components of watershed hydrology. Furthermore, we synthesized complex scientific data to discern historical trends and make predictions of various future scenarios. Landscape simulation models are useful tools for understanding hydrologic processes and predicting the corresponding impacts of landscape intervention actions. However—from a landscape management perspective—models are only useful if they provide relevant outputs that can inform policy trade-offs and decision-making. While the scientific information, technical evidence, and inferences employed in models must be credible and robust, they must also reflect multiple relevant social, economic, and policy perspectives. Therefore, developing a useful model as a boundary object to connect the scientific information to policy decisions and investment needs requires collaborative efforts among scientists and stakeholders [12].

The synthesis modeling framework engaged stakeholders from project onset to direct the overall model framework and selection of intervention options, in addition to providing input data, scientific and management feedback on the model details such as the computational structure and framing of model outputs. Ultimately, four models were synthesized to provide inputs to a financial model for the CWSS that overlays full-lifecycle costing and projected economic benefits for nature-based solutions investment scenarios to estimate the economic cost of a drought [13]. The Resource Investment Optimization System (RIOS) was used to define the optimized scenario for maximizing infiltration and baseflow. The Fog Interception for the Enhancement of Streamflow in Tropical Areas (FIESTA) was used to incorporate a fog water input in the watershed. Finally, the Hydrologic Modeling System from the Hydrologic Engineering Center (HEC-HMS) and the Soil and Water Assessment Tool (SWAT) were used to estimate the NbS effects on watershed hydrology of different landscape scenarios.

The objective of this paper is to demonstrate the synthesis modeling framework to simulate impacts of NbS scenarios on hydrological components and water availability, including the description of model algorithms, assumptions, input data, calibration, validation, and results.

## 2. Methods

In this section, we define the landscape NbS counterfactual scenarios, the climatic and hydrological inputs, the geospatial model inputs and parameters, as well as the main algorithms and assumptions of the hydrological models (HEC-HMS and SWAT) (Figure 2).

### 2.1. Landscape NbS Counterfactual Scenarios Definition

The Landscape NbS counterfactual scenarios were developed based on multiple lines of social and environmental data. First a Land Use/Land Cover (LULC) change analysis [5] was conducted using a set of historical LULC images for years 1985 through 2018 from Annual Mapping Project for Land Use and Land Cover in Brazil [14]. Second, possible intervention sites in riparian buffer zones were mapped following the definitions of the Brazilian Forest Code, using shapefiles of rivers and water bodies from Brazilian Institute of Geography and Statistics (IBGE) and National Water Agency (ANA), municipality and land tenure limits from SICAR (National Rural Environmental Registry System) database,

and applying the methodology of processing data described in [15]. In order to allow an initial diagnosis and to identify the areas and activities that will have the greatest impact on ecosystem services [16] RIOS model was executed given the CWSS geophysical characteristics (climate, topography, soil type, and underlying geology). RIOS is a model developed by the Natural Capital Project [17] to prioritize the areas, activities, and resources that can be used in NbS investment projects. The model makes it possible to conduct a spatially distributed analysis at pixel level (in this case, 30 m), with the location of the possible priority areas for intervention. The geospatial RIOS results were used to develop a customized scenario that prioritizes areas for infiltration and baseflow (Figure 3c).

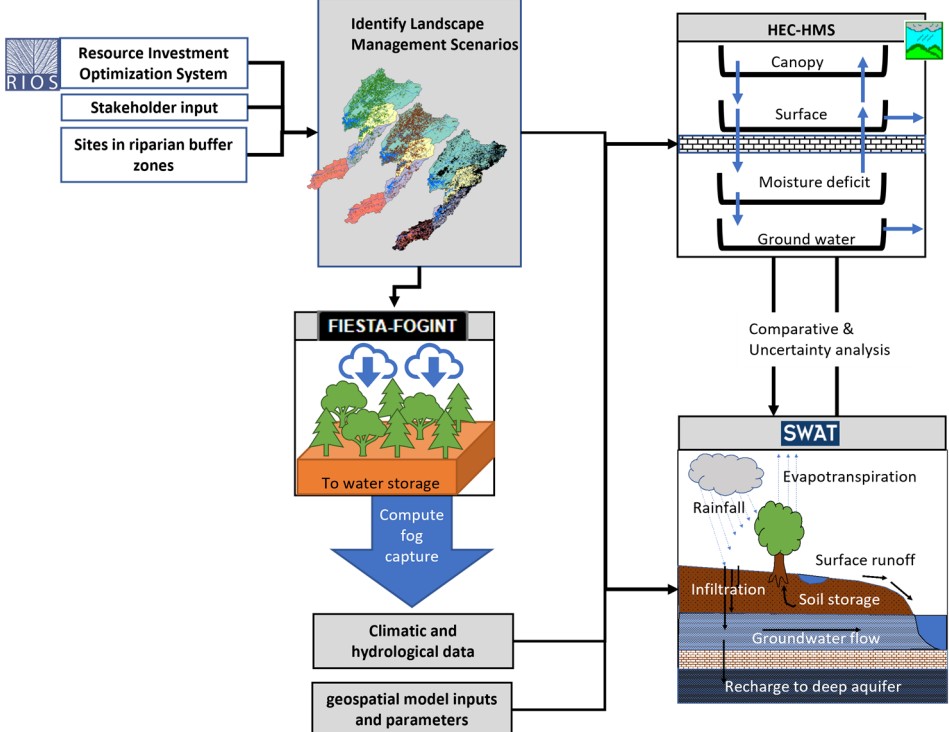

**Figure 2.** Methods steps and integration of multi-model framework (adopted Cho et al., 2023 [5]).

Finally, the scenarios selected for the impact evaluation consist of different spatially explicit allocations of forest, riparian, and agricultural intervention options. In addition to the baselines for each model, which reflect the watershed conditions in the year 2018, six intervention options were considered:

1. Minimum intervention (MI): Considers the restoration of degraded riparian buffer zones according to the minimum standards defined by the Brazilian Forest Code.
2. Enhanced intervention (EI): Follows the same logic as the minimum intervention scenario but considers the restoration of larger riparian buffers.
3. RIOS customized (RIOS): Prioritizes areas defined to maximize infiltration and baseflow given the CWSS geophysical characteristics.
4. Agricultural Best Management Practices (BMP): Considers the deployment of terracing and implementation of sediment watersheds on pastureland.
5. Pasture to Forest (PtF): Assumes the entire pasture landcover is converted to native forest.
6. Forest to Pasture (FtP): Assumes the entire native forest is converted to pasture.

The two later scenarios are purely hypothetical and serve to test the extreme conditions, in which the entirety of the pasture is converted to forest, and vice versa. Forest Code is interpreted and mapped over private and public lands to define the 'minimum' and 'enhanced' scenarios. In these two scenarios, all pasture areas that are under São Paulo State Water Utility Company (Companhia de Saneamento Básico do Estado de São

Paulo/SABESP) land tenure have been included in the restoration area. Figure 3 shows the total extent (%) and spatial distribution of NbS in each intervention scenario.

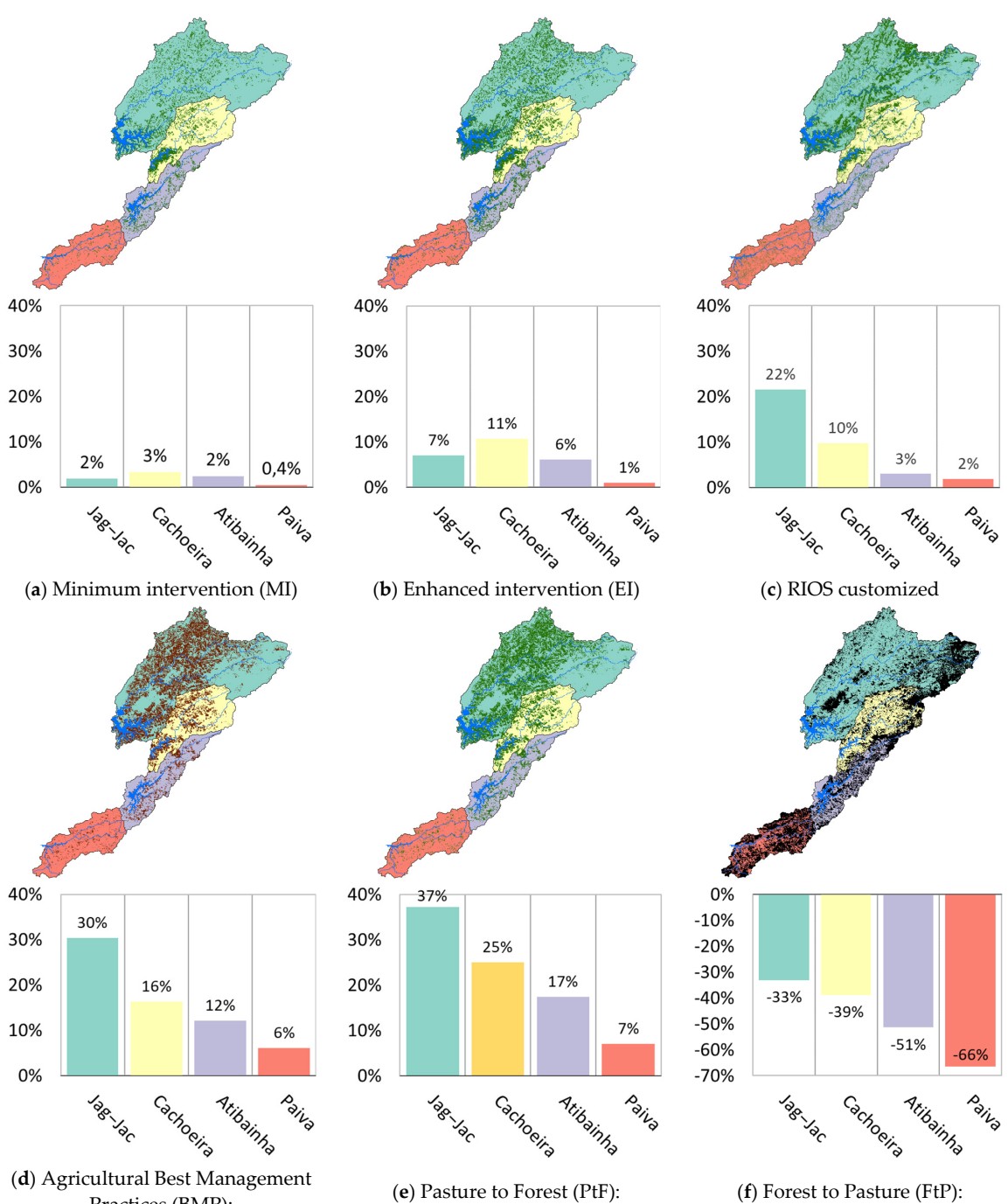

**Figure 3.** Landscape NbS counterfactual scenarios and scenario cover percent in each system.

### 2.2. Climatic and Hydrological Input

Input data for both hydrological models (SWAT and HEC-HMS) include daily precipitation from several rain gauge stations from *Sistema Nacional de Informações sobre Recursos Hídricos* (SNIRH) and *Instituto Nacional de Meteorologia* (INMET), across the watershed (Camanducaia, São Francisco Xavier, Vargem, Crioulos, Nazaré Paulista, Morro Grande, Guarau e Guirra) starting from 1 January 1980 to 31 January 2018. Likewise, the models were calibrated using hydrological monitoring data from *Companhia de Saneamento Básico do Estado de São Paulo* (Sabesp) and *Departamento de Águas e Energia Elétrica* (DAEE). Daily

reservoir inflow data (cms) were collected at the individual reservoirs (Jaguari-Jacarei, Cachoeira, Atibainha, and Paiva Castro) from 1 January 1984 to 31 December 2019, and at the stream gage on the Jaguari River (F25B Rio Jaguari) from 1 January 2008 to 31 January 2018. For the calibration step, daily flow data between 1 January 2004 and 31 May 2015 were selected, and for the validation of the model fit the period of 1 June 2015 to 1 January 2018 was selected. These choices were based on data availability.

The forest captures cloud water that could otherwise pass-through watersheds, providing additional water inputs to the systems. The additional water could be important, especially in regions characterized by dry season of little or no rain events. During dry season, fog contributions can be significant for biological processes and form an important part of the hydrological cycle in the forest. The occurrence of fog capture depends on locally specific micro-climatic and landscape relief preconditions such as temperature, altitude, prevailing wind currents, relief orientation, land cover, and season. FIESTA is a hydrological model that quantifies hydrological fluxes, including those contributed by fog interception in a watershed and is used to quantify the potential water balance gains with land use/land cover conversion. We used FIESTA to estimate total potential fog capture (mm/year) of the complete forest restoration across the CWSS at 1° resolution as the basis for evaluating restoration effects on the comprehensive precipitation input in the watershed (Figure 4). The model estimates fog capture based on cloud presence and considers spatial inputs of other climatic variables (wind, temperature, atmospheric moisture content, etc.), land cover, and topography [18]. These estimates are incorporated in the hydrological models in simulating long-term water balance for different landscape management scenarios.

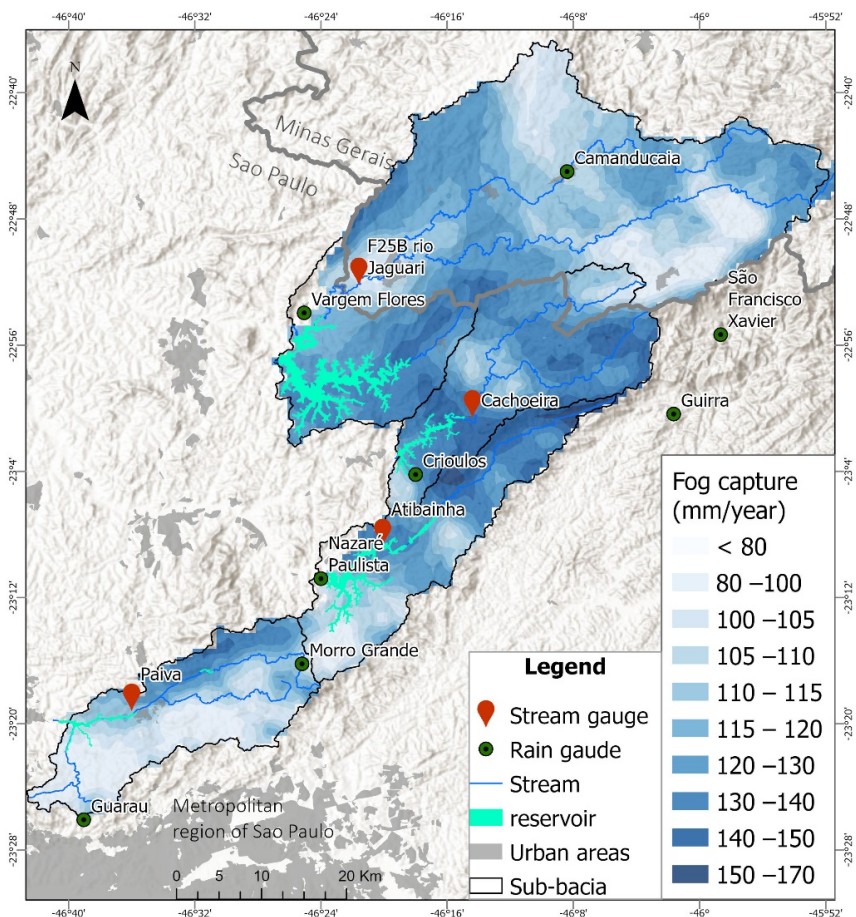

**Figure 4.** Monitoring station across the CWSS and FIESTA estimated fog capture potential with forest cover.

### 2.3. Geospatial Model Inputs and Parameters

Spatial input data include multiple data sources, from soil mapping, high-resolution topography to various remotely sensed data, and are used to parameterize the model on canopy interception and storage capacity, surface depression, surface runoff transformation, infiltration capacity, soil storage capacity, and subsurface flows.

- Land Use, Land Cover: For baseline, the land cover classes classified from Landsat images from year 2018, collection 5 from Annual Mapping Project for Land Use and Land Cover in Brazil [14] was used. The landscape intervention scenarios were developed from the baseline map of the watershed with various NbS allocations (Figure 3).
- Impervious surface and surface storage capacity: Surface parameter values include impervious surface and surface storage capacity. Impervious surface coverage values were estimated using LULC 2018 and the existing values provided by Sabesp. The surface depression storage values depend on both LULC2018 and the topography.
- Canopy storage input: Canopy storage depends on vegetation cover, for which we used the leaf area index (LAI) as a proxy for estimation with different land cover categories. LAI is calculated using Sentinel 2 data from 2019 processed on the Sentinel Application Platform (SNAP) [19,20]. To calculate the canopy interception, a modified Gash method was applied [21–23].
- Crop coefficient (Kc): The crop coefficient could be calculated as a ratio between ET and potential evapotranspiration (PET) (i.e., a measure of water extraction efficiency from soil). ET values were derived from MODIS observation from 2009 to 2019 using Google Earth Engine, and PET values are derived from CGIAR Aridity database and 100 years of temperature data. An independent study by [24] reported similar values of crop coefficient in a hydrologic modeling study for the Grande River Basin in Southern Minas Gerais State, Brazil.
- Soil classes: A map of soil classification was used from Embrapa to SWAT model, and different raster of soil maps from sentinel and google engine were used to parametrize both hydrologic simulation models.
- Soil parameter: Soil parameters values include maximum infiltration rate, soil storage, tension storage, percolation rate, groundwater storage, and groundwater coefficient. These parameters depend on soil properties and land cover/land use. We used soil's physical properties (i.e., sand, clay, and organic matter content, and soil depth) to calculate hydraulic conductivity, infiltration, soil saturation, and soil storage.

### 2.4. Hydrologic Simulation Model Development, Algorithms, and Assumptions

2.4.1. Hydrologic Modeling System of Hydrologic Engineering Center (HEC-HMS)

HEC-HMS originally used by Sabesp was set up to simulate event-based (~12–24 h) surface runoff given precipitation data input. During the February 2020 workshop with Sabesp, a process of converting the existing model for long-term continuous hydrologic simulations started—a model that simulates both wet and dry weather conditions, by adapting the soil moisture accounting (SMA) algorithm. We developed four independent models to represent the watersheds draining to corresponding individual reservoirs. Subsequently, we identified data needed for long-term hydrologic simulation, which have been supplied by Sabesp. When the long-term hydrologic simulation model set up was completed, we worked with the Sabesp's engineering team to evaluate the model framework, input data, and outputs.

Soil moisture accounting (SMA) model simulates storage and movement of water at each sub-basin across the watershed. Given hydrometeorological data, the model computes watershed surface runoff, losses through evapotranspiration (ET), infiltration, groundwater flow, and deep percolation [23]. The SMA algorithm keeps track of water balance in the form of surface runoff, evapotranspiration, infiltration, and lateral and base flows through a series of storage components: The canopy-interception storage represents precipitation intercepted by trees, shrubs, and grasses. Water in canopy storage is held until it is removed



by evaporation (loss of water that is no longer a part of the downstream water budget). The surface-interception storage represents precipitation, which is not intercepted by canopy (i.e., throughfall) and not infiltrated to soil column, held in shallow surface depressions. The precipitation volume in excess of the available surface storage volume becomes surface runoff and contributes to sub-watershed water yield. The soil-profile storage represents water stored in the top layer of the soil in the upper zone and tension zone. Inflow is infiltration of precipitation throughfall, and outflows include percolation to groundwater layers from upper and tension zones and loss through plant uptake and ET from tension zone. Finally, the groundwater layers represent horizontal interflow processes, which are modeled as simple linear reservoirs. Inflow is the water percolating from the soil profile and outflow is groundwater flow contribution to sub-watershed water yield.

SMA model parameters must be determined by calibration with observed data. Calibration parameters include soil storage, tension storage, percolation, groundwater storage, and base flow coefficients [25]. These parameters are less likely to be immediately responsive to the restoration actions. The NbS scenarios are simulated by altering parameters that respond to conservation actions, based on literature and experimental data, including canopy interception, crop coefficient, surface depression and storage, maximum infiltration rate, and impervious surface percentage. Calibration routine used the deterministic optimization routine on HEC-HMS to improve the model performance. Beginning with initial parameter estimates, the optimization routine adjusts the calibration parameter values until the simulated results match the observed hydrograph as closely as possible. In the optimization routine, root mean square error (RMSE) was used to minimize the average difference between observed and simulated, with larger weights on larger errors. Table 1 shows the fitness of the calibrated and validated models.

**Table 1.** HEC-HMS calibration and validation results.

| Watershed | Observed Flow | | Simulated Results | | Objective Functions | | |
|---|---|---|---|---|---|---|---|
| | Peak Discharge (cms) | Water Yield (mm) | Peak Discharge (cms) | Water Yield (mm) | RMSE Std. Dev. | Nash-Sutcliffe | Percent Bias |
| Calibration (2 January 2004–31 May 2015) (F25BT: 1 January 2008–31 January 2018) | | | | | | | |
| Jag-Jac | 273 | 6069 | 95 | 6515 | 0.60 | 0.67 | 7% |
| Jag-Jac (F25BT) * | 358 | 4525 | 98 | 5827 | 0.70 | 0.51 | 29% |
| Cachoeira | 93 | 5024 | 64 | 5852 | 0.60 | 0.61 | 17% |
| Atibainha | 65 | 5405 | 41 | 5887 | 0.60 | 0.61 | 9% |
| Paiva-Juq | 68 | 5453 | 32 | 4362 | 0.80 | 0.33 | −20% |
| Validation (1 June 2015–31 January 2018) | | | | | | | |
| Jag-Jac | 154 | 1247 | 95 | 1593 | 0.70 | 0.52 | 28% |
| Cachoeira | 52 | 914 | 27 | 1191 | 0.80 | 0.40 | 30% |
| Atibainha | 47 | 905 | 27 | 1152 | 0.70 | 0.57 | 27% |
| Paiva-Juq | 70 | 1023 | 48 | 1036 | 1.00 | 0.04 | 7% |

Note: * At Jaguari-Jacarei (Jag-Jac), there were two independent stream flow observations (stream gages and reservoir inflows) available over different periods, and both data are used to evaluate the model.

### 2.4.2. Soil and Water Assessment Tool (SWAT):

The Soil and Water Assessment Tool (SWAT) model is a semi-distributed and continuous model, developed to simulate the impact from LULC modifications on the hydrology at different scales. It incorporates a set of equations that represent in a simplified way water, sediments and nutrients move across a watershed [26]. In SWAT the hydrologic cycle is estimated considering the variations between the water inputs and outputs in the watershed, in which soil storage parameters are used to evaluate the impact of changes is ambient water storage with the landscape NbS scenarios. The water that enters the soil column takes different pathways: water could be evaporated, taken up by plants, percolated,

and recharge the deeper aquifer; or the water could move laterally and contribute with the stream. The representation of these balance is Equation (1):

$$SW_t = SW_o + \sum_{i=1}^{t} \left( R_{day} - Q_{surf} - E_a - W_{seep} - Q_{gw} \right) \tag{1}$$

where: $SW_t$ = final soil water content in mm; $SW_o$ = initial soil water content (mm on day $i$); $R_{day}$ = precipitation (mm on day $i$); $Q_{surf}$ = surface runoff (mm on day $i$); $E_a$ = evapotranspiration (mm on day $i$); $W_{seep}$ = water entering the vadose zone from soil profile (mm on day $i$); $Q_{gw}$ = amount of return flow (mm on day $i$); $t$ = time in days.

In addition to soil water storage, an important part of the balance that was evaluated in this project is the groundwater contributions. Although SWAT has limitations in the groundwater component, like every model, it has simplifications that may not include all the specificities of the aquifer's layers and interactions between different hydrogeologic structures. However, it was important to compare the behavior of this plot of water in each of the scenarios as a part of a comprehensive water balance analysis. SWAT represents groundwater using three reservoirs [26]: (i) soil layers where water is stored and re-distributed: this water is available for plant uptake, evaporation or to recharge the shallow aquifer; (ii) the shallow aquifer is located below the soil profile and is separated by a 'vadose zone' which receives water percolated from the lowest layer of the soil. The shallow aquifer can either discharge into the nearest stream as groundwater contribution to baseflow, or percolate to contribute to the deep aquifer; (iii) the deep aquifer for SWAT represents contributions to streamflow outside of the analyzed watershed. In the case of CWSS, the deep aquifer represents contributions to either another system or water available for pumping via deep wells. This is noteworthy because the CWSS is located upstream of a significant watershed that relies on the recharge contributions to Brazilian aquifers. A high percentage of water in the CWSS is stored in the soil and aquifer layers, and as a result modelling outputs for field interventions show up disproportionately in soil storage component rather than surface flows.

The calibration and sensitivity analysis of the SWAT model was performed with automatic program SWAT-CUP, using the SUFI2 (Sequential Uncertainty Fitting) method. Latin hypercube method is used in the calibration process within the defined intervals for calibration and the number of simulations to be performed [27]. For calibration, the program was run exhaustively, analyzing the sensitivity of different parameters related to soil, groundwater, subsurface flux, interception process, first together and then separately for each process. After the analysis, the most sensitive parameters for calibration were selected, including the saturated hydraulic conductivity (SOL_K), moist bulk density (SOL_BD), the available water capacity of the soil (SOL_AWC), baseflow alpha factor for bank storage (ALPHA_BNK), Manning's "n" value (CH_N2) and groundwater delay (GW_DELAY) [28]. To simulate the differences in soil parameters (e.g., porosity, infiltration rate, soil storage, tension storage, percolation rate, groundwater storage, and groundwater coefficient), they were calibrated by the type of soil (Table 2).

**Table 2.** SWAT most significant models' parameters.

| Parameters | Description | Initial Values | Calibrated Values |
|---|---|---|---|
| SOL_K for soil type 1 | Saturated hydraulic Conductivity for soil type1 | 0–2000 | 28.00 |
| SOL_K for soil type 2 | Saturated hydraulic Conductivity for soil type2 | 0–2000 | 25.00 |
| SOL_K for soil type 3 | Saturated hydraulic Conductivity for soil type3 | 0–2000 | 8.00 |
| SOL_K for soil type 4 | Saturated hydraulic Conductivity for soil type4 | 0–2000 | 9.00 |
| SOL_AWC soil type 1 | Available water capacity of the soil for soil type 1 | 0–1 | 0.35 |
| SOL_AWC soil type 2 | Available water capacity of the soil for soil type 2 | 0–1 | 0.15 |
| SOL_AWC soil type 3 | Available water capacity of the soil for soil type 3 | 0–1 | 0.10 |
| SOL_AWC soil type 4 | Available water capacity of the soil for soil type 4 | 0–1 | 0.38 |
| ALPHA_BNK | Bank storage | 0–1 | 0.59 |
| CH_N2 | Manning's "n" value | 0–0.3 | 0.24 |
| GW_DELAY | Groundwater delay | 0–500 | 94.0 |

To evaluate the SWAT performance, the following objective functions were used: Nash–Sutcliffe coefficient (NSE), the coefficient of determination ($R^2$) and the percent bias (BIAS) (Table 3).

**Table 3.** SWAT calibration and validation results.

| Watershed | Observed Flow | | Simulated Results | | Objective Functions | | |
|---|---|---|---|---|---|---|---|
| | Peak Discharge (cms) | Water Yield (mm) | Peak Discharge (cms) | Water Yield (mm) | Coefficient of Determination $R^2$ | Nash-Sutcliffe | Percent Bias |
| Calibration period (1 January 2004–31 May 2015) | | | | | | | |
| Jag-Jac | 273 | 6069 | 218 | 5710 | 0.73 | 0.7 | 5% |
| Cachoeira | 93 | 5024 | 51 | 5755 | 0.62 | 0.56 | −15% |
| Atibainha | 65 | 5405 | 41 | 4024 | 0.37 | 0.33 | 22% |
| Paiva-Juq | 68 | 5453 | 50 | 4432 | 0.31 | 0.22 | 19% |
| Validation period (1 June 2015–1 January 2018) | | | | | | | |
| Jag-Jac | 154 | 1168 | 146 | 1219 | 0.73 | 0.72 | −6% |
| Cachoeira | 52 | 878 | 37 | 1131 | 0.6 | 0.36 | −33% |
| Atibainha | 47 | 820 | 14 | 638 | 0.35 | 0.33 | 18% |
| Paiva-Juq | 70 | 1023 | 63 | 1010 | 0.19 | 0.04 | 2% |

## 3. Results and Discussion

By applying HEC-HMS and SWAT, the effects of landscape restoration on multiple components of watershed hydrology were simulated to better: (i) assess optimal locations for NbS interventions, (ii) predict changes in groundwater and surface water balance, and (iii) simulate water availability and ambient storage. These independent hydrologic models are utilized to understand long-term watershed hydrology and to simulate the effects of the landscape intervention scenarios. These models are characterized by different levels of complexity and functionality, and the pros and cons associated with them. SWAT considers different land uses and soils and within a sub-watershed with higher spatial resolution to simulate surface runoff, subsurface flows and storage, along with water quality metrics. As a result, SWAT requires considerable expertise and time in model development and interpretation. On the other hand, HEC-HMS has relatively small number of parameters within a simpler model structure that can capture the general watershed hydrology, illustrate watershed processes, and provide immediate feedback—with a relatively short model run time—on the watershed response to different landscape intervention scenarios. Both models are used to simulate the long-term hydrology in the study site. In general, compared to observed data, calibrated HEC-HMS provided a better response for average flows, while SWAT resulted in a better correlation with extremes of the hydrograph—peaks and low flows (Figure 5a,b). These models provide two lines of evidence to infer the long-term hydrologic impacts of different landscape intervention scenarios.

The simulation outputs from 1 January 1988–1 January 2018 inform changes in the rain-discharge relations and the overall long-term water balance in response to the landscape intervention scenarios, and highlight the implications of expanding distributed water storage through reforestation across the watershed on the overall water yield and its management.

In general, the model outputs indicate that the intervention options result in decline in extreme low flows and attenuate peak flow intensities with marginal improvements in the overall surface water yield. The SWAT model, when RIOS customized scenario is considered, indicates that the combination of surface water flows and ground water to stream flows provide an overall increase and thus, the potential for long-term watershed replenishment capacity facilitated by NbS. Water quality improvements with reductions in the sediment and nutrient loads are also observed.

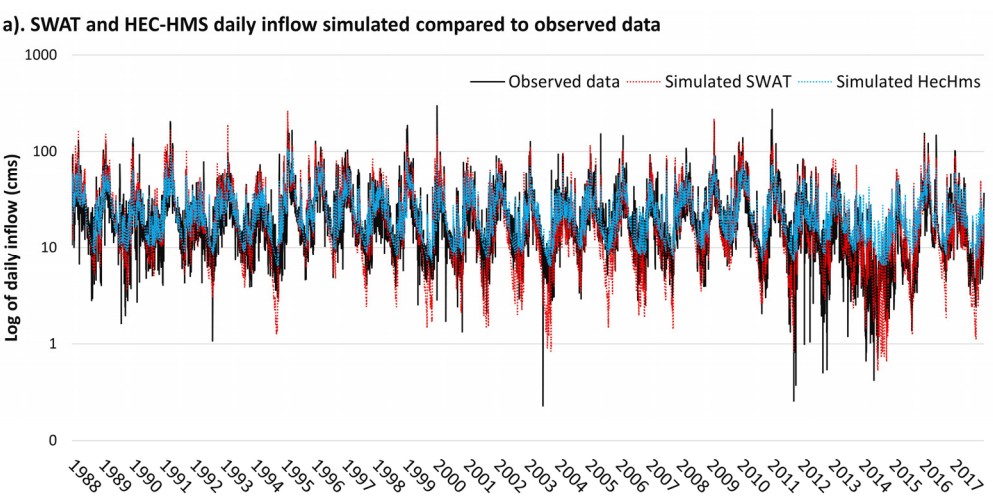

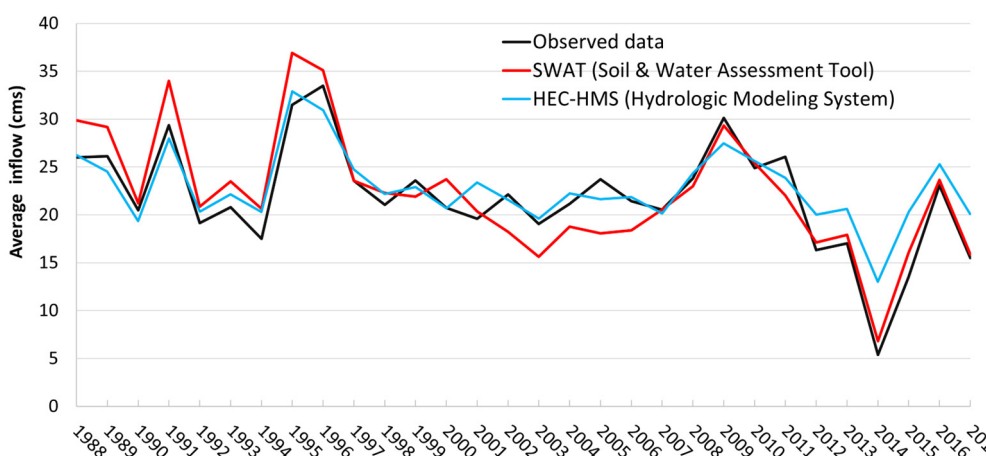

**Figure 5.** Comparation between calibrated model outputs and observed data, in (**a**) daily data, and (**b**) year average data.

### 3.1. HEC-HMS Biophysical Simulation Model Output Summary

Landscape restoration scenarios effects on watershed hydrology are simulated using the HEC-HMS models calibrated and validated to the present land use/land cover condition. The long-term hydrologic simulations are executed from 1 January 1987 to 1 January 2018, and the model outputs are evaluated over the 30-year period from 1 January 1988 to 1 January 2018. We compare the scenario outputs, the extreme cases and the baseline using some indicators, such as water yield, peak, and extreme low flows. Then, the average monthly changes in water budget for each scenario were analyzed.

3.1.1. Mean Annual Changes in Water Budget over the Entire Simulation Period

In general, the HEC-HMS long-term hydrologic simulation outputs indicate that the restoration activities result in declines in the extreme low flows and peak floods with marginal improvements in the annual water yields (WY) (Figure 6a). Among all scenarios, RIOS generally performs best to improve water quantity benefits in all watersheds. Among all watersheds, Jaguari-Jacarei shows most improvement with RIOS scenario. Other scenarios based on Forest code (MI and EI) show marginal improvements. The extreme case PtF, while showing only slight improvements in overall increase in WY, resulted in relatively larger improvements in mitigating extreme low flows and peak flows. Case FtP resulted in larger surface runoff, as the overall infiltration and ET rates decline with forest conversion to pasture, and an overall increase in WY while both low flow and peak flow events increase.

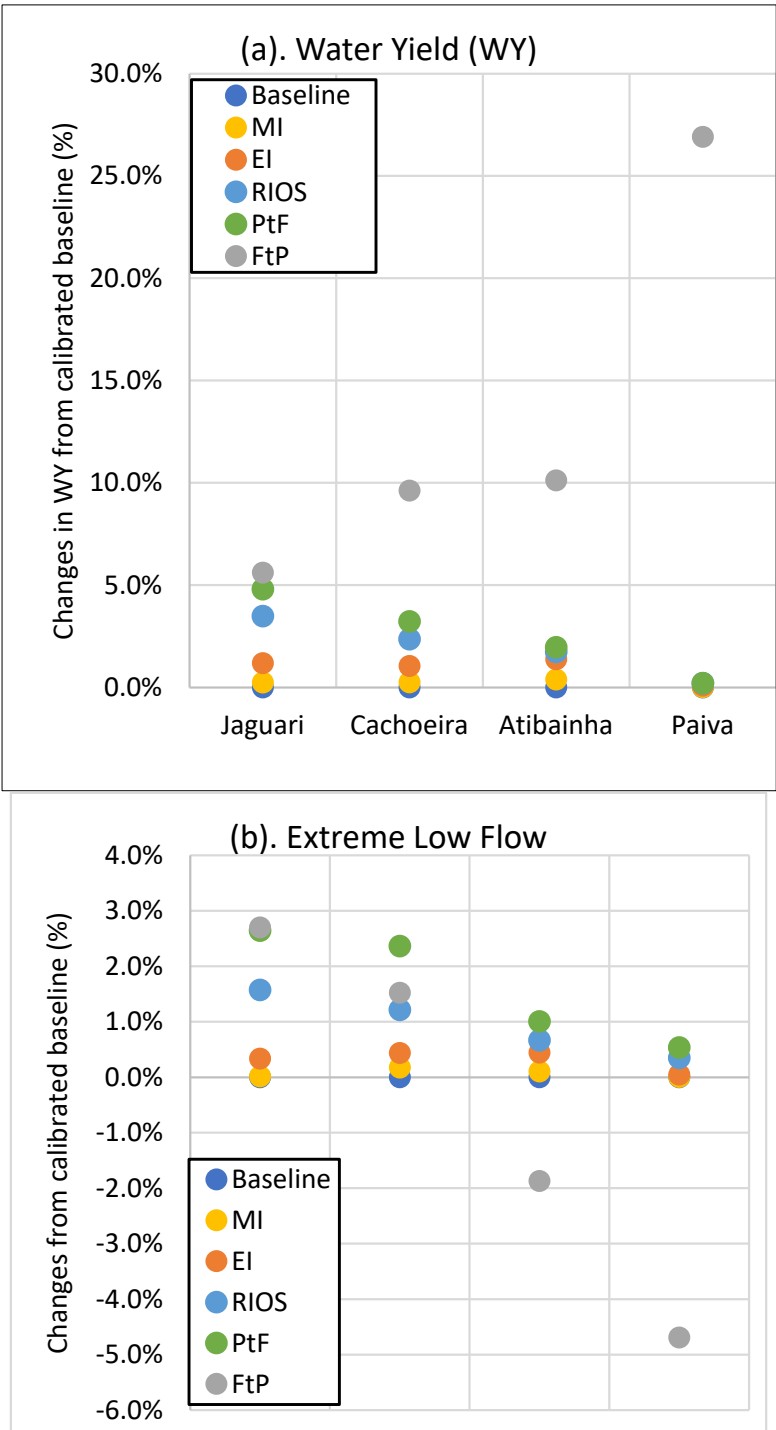

**Figure 6.** *Cont.*

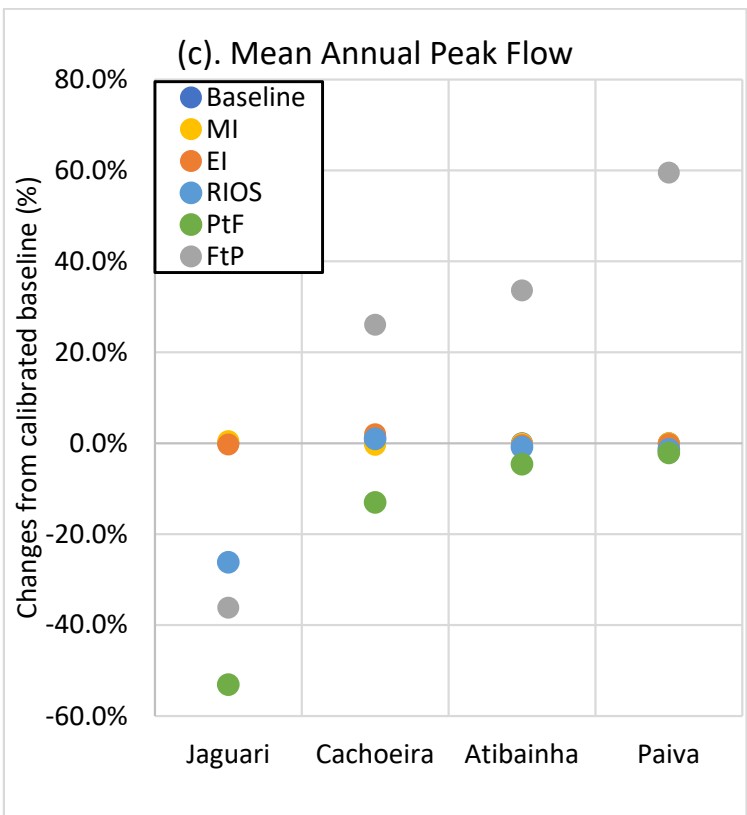

**Figure 6.** Long-term average changes in water quantity measures derived from the 30-year hydrologic simulation in Jaguari-Jacarei (Jag/Jac), Cachoeira, Atibahinha, and Paiva-Juqu with HEC-HMS (**a**) changes in the overall water yield, (**b**) changes in extreme low flows, and (**c**) changes in the mean annual peak flows.

### 3.1.2. Average Monthly Changes in Water Budget

Average monthly flows are calculated for the entire simulation period (1988–2018) and through the drought years (2013–2015). River discharge generally increases with the restoration scenarios throughout the years. The greatest increases are observed in Jaguari-Jacarei watershed, and the RIOS scenario resulted in the largest increase in river discharge in all watersheds, except in Atibainha, in which EI scenario performed better. Though flows in the Cachoeira, Atibainha, and Paiva Castro are comparable, the restoration scenarios in Paiva Castro result in the smallest changes in the flow since it has least amount of restoration area allocation (Figure 7a). There are greater increases in river discharge during the drought period (2013–2015) compared to the entire simulation period, indicating that the restoration scenarios have the potential to deliver water when it is scarce, particularly during the dry season (Figure 7b).

HEC-HMS simulation outputs provide a general understanding of how the restoration scenarios affect water availability at the reservoirs. In the following section, the details of simulated water budgets using SWAT simulation outputs are examined.

### *3.2. SWAT Biophysical Simulation Model Output Summary*

### 3.2.1. Water Balance

SWAT model has the capacity to generate multiple elements of the water balance, including surface, subsuface, and groundwater flows. SWAT estimated daily soil water balance, given the computed water infiltration in the soil layers for every hydrologic response unit (HRU). Figure 8 indicates the main SWAT water balance components for the Jaguari baseline including precipitation, overland flow, lateral flow, groundwater contribution to base flow, soil water, and percolation.

(**a**)

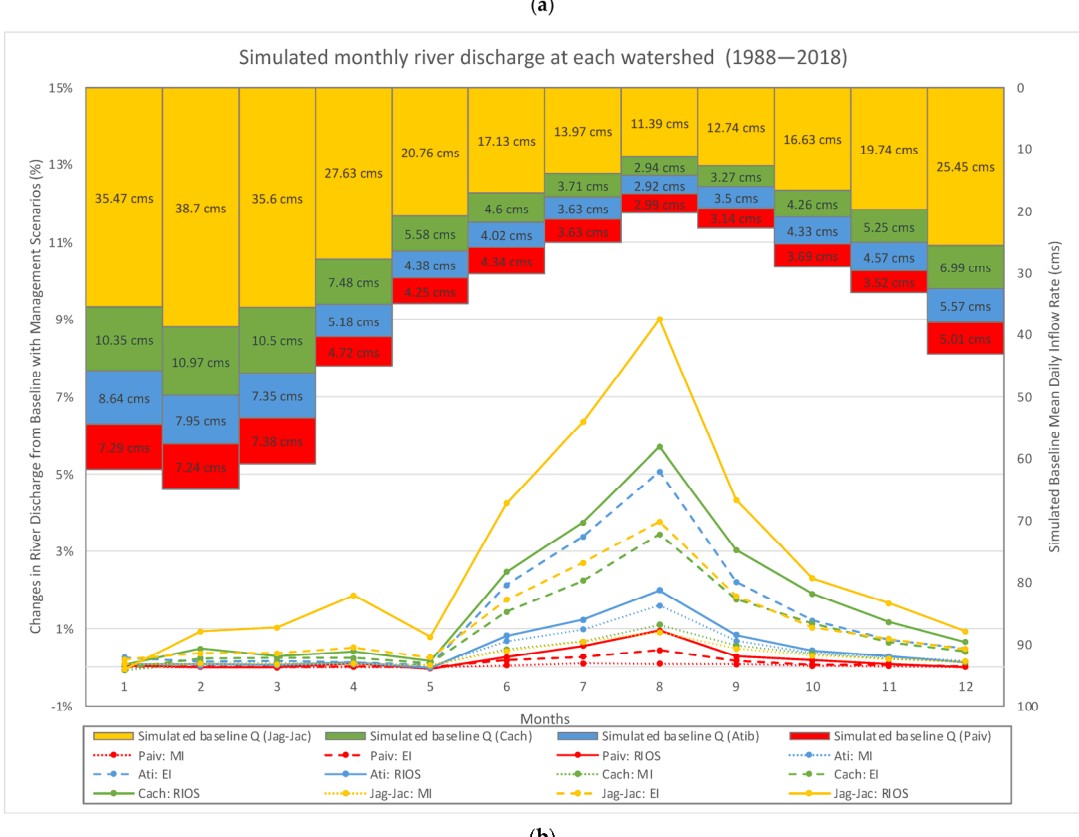

(**b**)

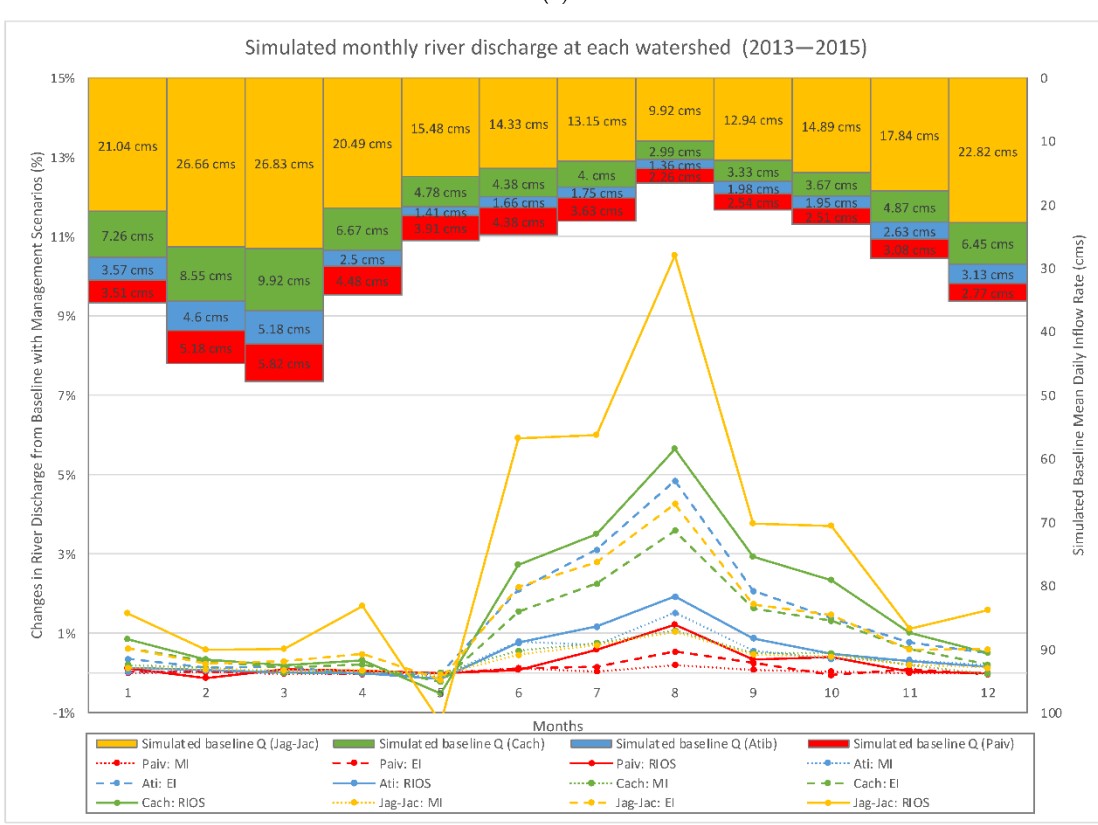

**Figure 7.** Average monthly river discharge and the changes with each scenario for (**a**) the entire simulation period, and (**b**) drought years.

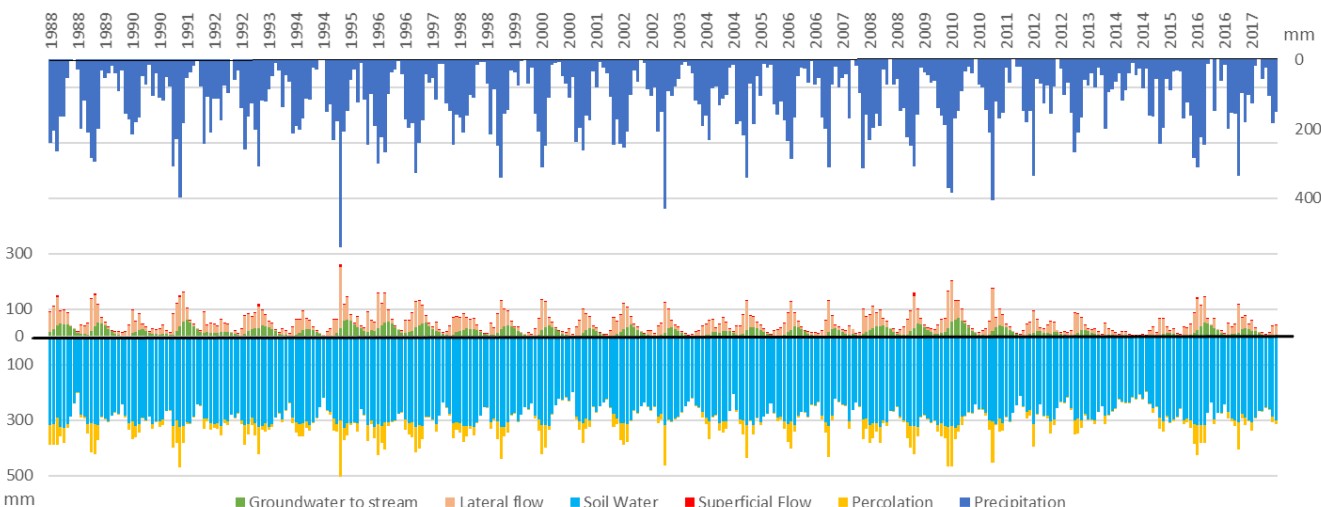

**Figure 8.** Water balance components (mm) in the SWAT outputs for Jaguari baseline.

Note that soil water storage also represents the cumulative water that is stored during the years of analysis, showing that NbS could help maintain water security and biodiversity not only for the source watersheds where the Cantareira reservoirs are located, but also for neighboring rural and other municipalities through deep aquafer recharge. Furthermore, changes in this water storage could influence in the phreatic level, and in the movement lateral of water to the reservoir. The water balance components, groundwater recharge, and deep aquifer percolation were individually analyzed.

### 3.2.2. Groundwater Components

Model results show an increase in the recharge on average with NbS application, particularly when we compare the baseline and RIOS scenarios in the outlet basin (Figure 9a). In SWAT the landscape is divided into areas of recharge and discharge areas (Figure 9b) and the results could be significantly influenced by topography [26] where the water flow is directed toward the water table, or the stages of the reservoirs. Therefore, as the RIOS scenario maximizes the recharge to the aquifers through site prioritization, the most significant effect can be seen at the reservoir intake locations.

The bar chart in Figure 9d shows the behavior of all the sub watersheds in the crisis year of 2014 and 2015, where despite having a severe decrease in the amount of rain, the discharge watersheds have more water in the RIOS scenario compared to the baseline. These results support the hypothesis that NbS can contribute to overall balance, by increasing soil moisture content and associated aquifer recharge in the CWSS. Due to model limitations, it is impossible to predict how much of this water eventually will become part of the baseflow versus aquifer recharge; however, it is clear that NbS promotes overall recharge of the shallow and deep aquifers.

### 3.2.3. Soil Water Storage

On average, when considering all CWSS sub-watersheds (i.e., Figure 9c) SWAT model simulates increase in soil water content by 22% across the CWSS, and 27% increase in the discharge sub-watersheds (i.e., Figure 9b). Figure 10 shows significant increases in the water balance in all months, as related to the 'Soil Water' component in SWAT.

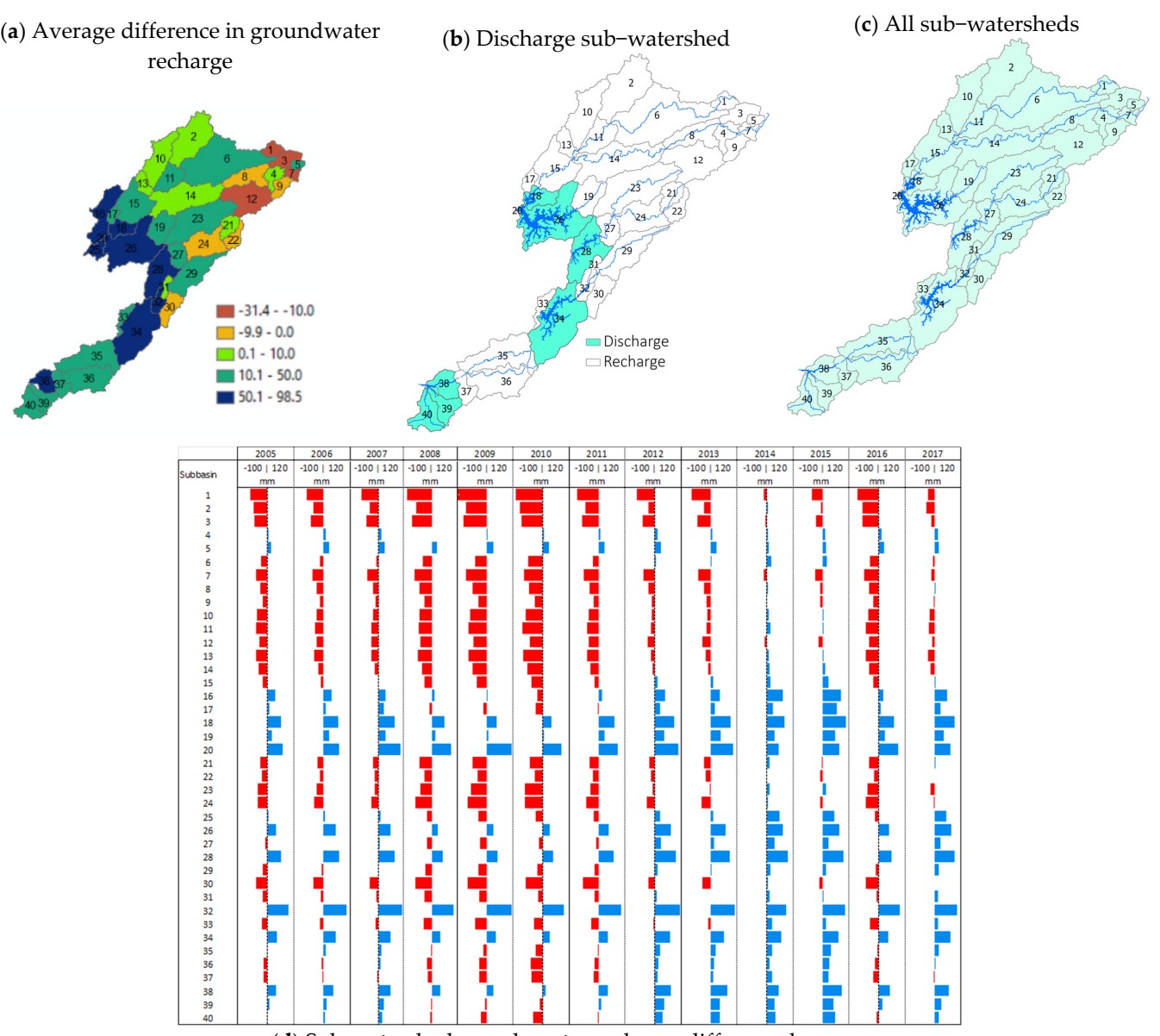

(**d**) Sub-watershed groudnwater recharge difference by year.

**Figure 9.** Difference between RIOS and BASELINE for groundwater recharge average for 2014–2015 (**a**); all CWSS contributing sub-watersheds map (**b**) discharge sub-watersheds only (**c**); Yearly difference between RIOS and BASELINE groundwater recharge by watershed (**d**).

### 3.2.4. Water Flow Components

By examining the combination of the water flow components—namely surface flows and groundwater to stream flows—SWAT model outputs indicate an average 33% increase, or 206 hm$^3$/year, when comparing RIOS customized against the baseline scenario and when all sub-watersheds in the CWSS are considered (Figure 11). In the discharge sub-watersheds, the increase in the water flow components was more significant with a 58% average increase (or 341 hm$^3$/year).

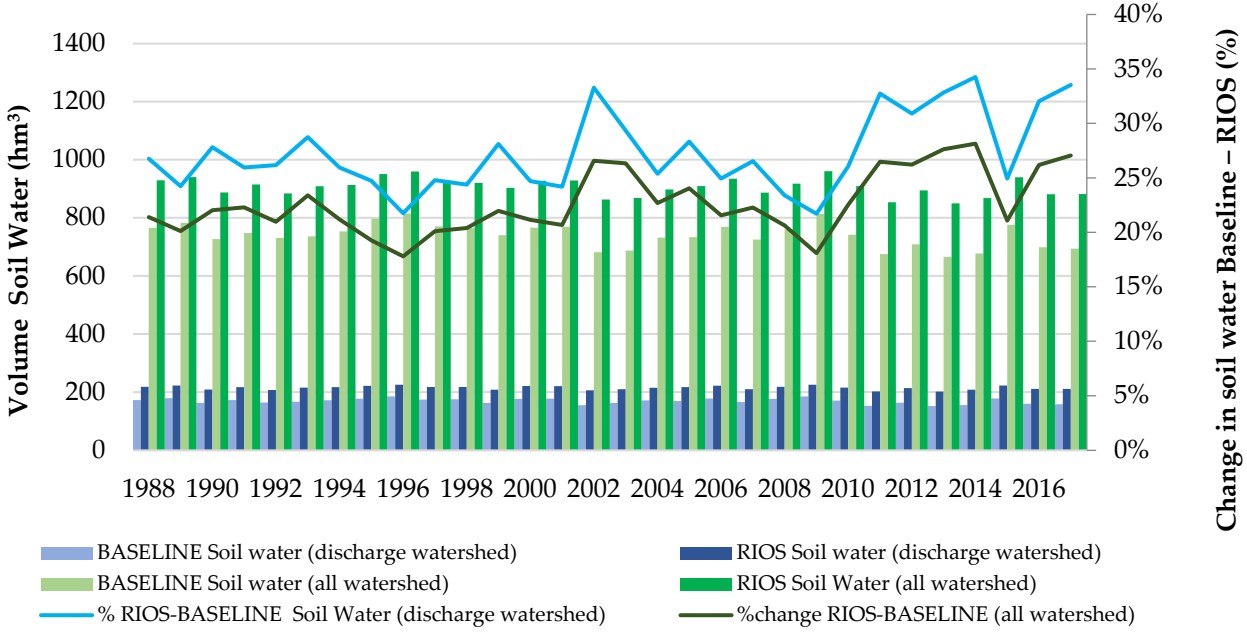

**Figure 10.** Annual CWSS percentage water balance contribution comparing RIOS to baseline scenario (1988–2017).

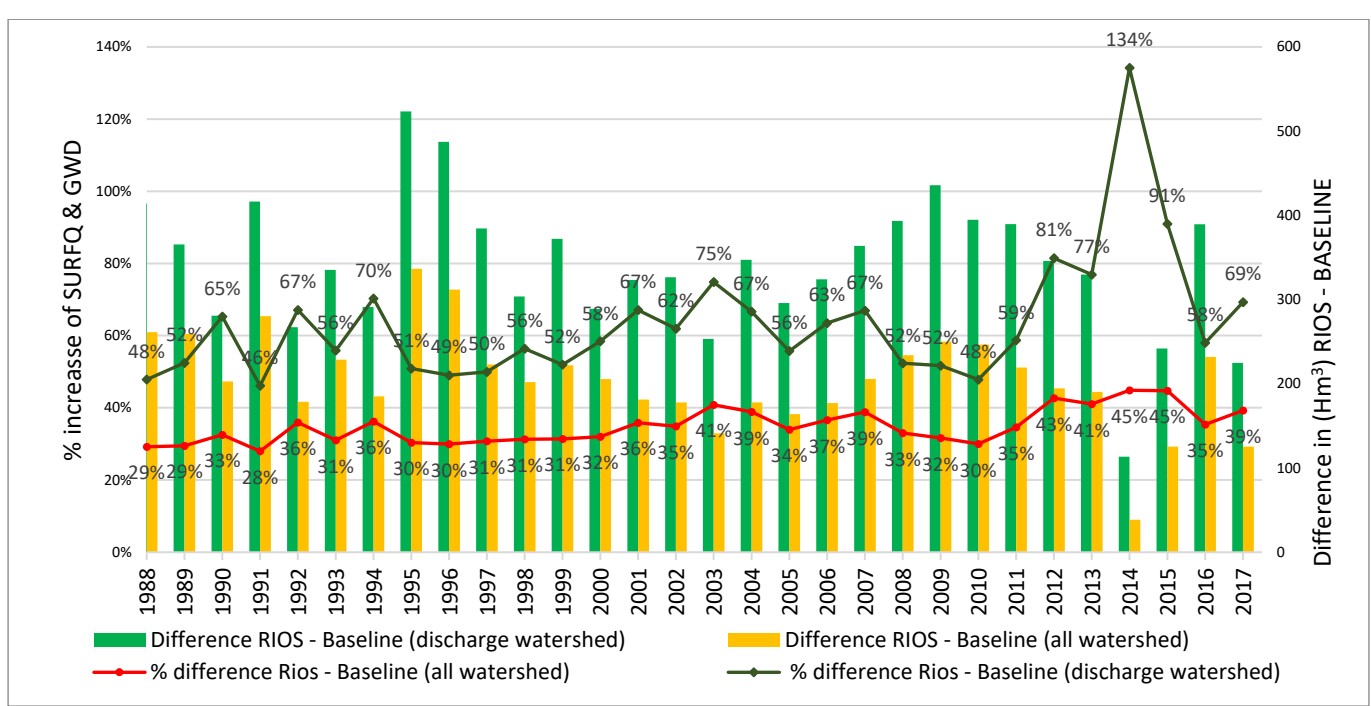

**Figure 11.** Annual CWSS difference in water flow components (∆ and %) comparing RIOS to baseline scenario (1988–2017).

These two water components provide benefits during periods of low water availability such as the 2013–2015 drought or other periods of water scarcity. A significant increase in these flow components is observed in the 2014–2015 drought period. This trend reflects the benefits of adopting NbS interventions in the CWSS—particularly during extreme climate conditions—providing resilience to the source watershed system.

In general, the two models show that there are benefits associated with the implementation of NbS. Obviously, due to the complexity of the hydrological system and its cycle, it is possible that when observing these benefits only in the surface flow for example, the

expected benefit may have low significance. However, we must take into consideration that the water in the basin is not just what we see, as signaled by the UN in the World Water Development report for groundwater: Making the invisible visible [29], which highlights that groundwater accounts for approximately 99% of all liquid fresh water on Earth. The importance of groundwater component to the overall watershed hydrology is also confirmed by [30], who identified the hydrogeological structures of Piracicaba-Capivari-Jundiaí (PCJ) (CWSS is part of PCJ system), and controlled the base flow processes in response to human activities and climatic variations. The study mentions that a major part of the total flow in the watershed depends on the contributions of the base flow (from between 40% and 75%). Therefore, the flow in the PCJ basins is vulnerable during the dry season and the different water uses could affect the water resilience, especially the surface water with shorter residence times and greater dependence upon precipitation.

## 4. Conclusions

A multi-model synthesis involved integration of landscape hydrologic simulation models, HEC-HMS and SWAT, along with FIESTA to estimate fog-capture potentials and RIOS to inform NbS allocation prioritization, the Brazilian Forest Code to develop various landscape intervention scenarios. Comparing different scales of NbS interventions scenarios, the study concludes that the areal extent and prioritization on intervention allocation according to the source watershed's geophysical conditions delivers the most significant improvement in water availability, especially during drought conditions. For example, the minimum and enhanced intervention scenarios resulted in marginal improvement in water availability, while the RIOS customized scenario resulted in the most significant improvement.

The results in the water dynamics indicated by the different hydrological models are complementary and corroborate with one another. The decreasing water availability trend in the 30-year observational data are reflected in both HEC-HMS and SWAT model's baseline outputs. With NbS allocation scenarios, models simulated the attenuation of peak flows and alleviation of extreme droughts. The SWAT model allowed an independent analysis of more complex water balance components including precipitation, lateral flow, groundwater contribution to base flow, soil water, and percolation. Modeling results showed that NbS contributes toward promoting water security by increasing water retention time and ambient storage in the CWSS, increasing the year-round soil and groundwater water volume, and improving the water recharge into streams during dry season, which can help to mitigate extreme drought conditions.

Implementation of NbS to increase ambient water storage and to attenuate surface flow is a relevant climate-change adaptation strategy to achieve source watershed resiliency in the face of increasing magnitude and frequency of extreme events. These model results clearly illustrate the "sponge effect" performed by the soil and groundwater storage by promoting infiltration through NbS allocation and subsequent increases in the overall flow to streams and aquifers. Such observations lead to an appropriate reference to the soil and groundwater components as the "invisible reservoir" of the CWSS, and its contribution to potential water benefits should be accounted for in building sustainable and resilient source watersheds.

**Author Contributions:** Conceptualization, J.R., C.K. and S.B.; Methodology, S.J.C. and E.A.A.; Software, E.A.A., S.J.C., J.L. and C.A.R.-P.; Validation, E.A.A. and S.J.C.; Formal analysis, E.A.A. and S.J.C.; Investigation, E.A.A., S.J.C., C.K., J.R. and H.B.; Data curation, E.A.A., S.J.C., J.L and H.B.; Writing—original draft preparation, E.A.A. and S.J.C.; Writing—review and editing, S.J.C., E.A.A., C.K. and B.S.C.; Visualization, E.A.A. and S.J.C.; Supervision, C.K. and S.B.; Project administration, C.K. and J.R.; Funding acquisition, S.B., J.R and C.K. All authors have read and agreed to the published version of the manuscript.

**Funding:** This research was funded by National Science Foundation (NSF) grant number DBI-1639145 and the Tinker Foundation. The APC was funded by the Tinker Foundation.

**Data Availability Statement:** Many of the data were acquired from public sources, including National Water and Sanitation Agency (ANA), and Brazilian Institute of Geography and Statistics. Some of the data were acquired directly from stakeholders and publications mentioned in the methods.

**Acknowledgments:** We thank Mara Ramos, Suely Matsuguma, Giovana Bevilacqua Frota, and Emerson Martins Moreira (Sabesp). We also thank our collaborators, including Rafael Barbieri and Suzanne Ozment (WRI) and Edenise Garcia, Maria Tereza Leite Montalvao and Arley Haley Faria (TNC) for their technical support and contributions, as well as SESYNC, and Tinker Foundation for the financial support to this study.

**Conflicts of Interest:** The authors declare no conflict of interest.

## Nomenclature

| | |
|---|---|
| alpha_bnk | Baseflow alpha factor for bank storage |
| ANA | National Water and Sanitation Agency |
| BIAS | Percent bias |
| BMP | Agricultural Best Management Practices |
| CGIAR | Consultative Group on International Agricultural Research |
| cms | Cubic meters by second |
| CWSS | Cantareira Water Supply System |
| DAEE | Department of Water and Electricity |
| EI | Enhanced intervention |
| FIESTA | Fog Interception for the Enhancement of Streamflow in Tropical Areas |
| FtP | Forest to Pasture |
| gw_delay | Groundwater delay |
| HEC-HMS | Hydrologic Modeling System from the Hydrologic Engineering Center |
| HRU | Hydrologic response unit |
| IBGE | Brazilian Institute of Geography and Statistics |
| INMET | National Institute of Meteorology |
| LAI | Leaf area index |
| LULC | Landuse/land cover |
| MI | Minimum intervention |
| mm | Millimeters |
| mm/y | Millimeters by year |
| n | Manning's "n" value |
| NbS | Nature-based solutions |
| NSE | Nash–Sutcliffe coefficient |
| ONU | United Nations Organization |
| PCJ | Piracicaba-Capivari-Jundiaí |
| PET | Potential evapotranspiration |
| PtF | Pasture to forest |
| RIOS | Resource Investment Optimization System |
| RMSE | Root mean square error |
| RMSP | Metropolitan Region of São Paulo |
| Sabesp | Basic Sanitation Company of the State of São Paulo |
| SICAR | National Rural Environmental Registry System |
| SMA | Soil moisture accounting |
| SNIRH | National Water Resources Information System |
| sol_awc | Available water capacity of the soil |
| sol_bd | Moist bulk density |
| sol_k | Saturated hydraulic conductivity |
| SUFI2 | Sequential uncertainty fitting |
| SWAT | Soil and Water Assessment Tool |
| WY | Annual water yields |

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
