# Peer review of "Biophysical Benefits Simulation Modeling Framework for Investments in Nature-Based Solutions in São Paulo, Brazil Water Supply System"

_water, doi:10.3390/w15040681_

Round 1
Reviewer 1 Report
The topic of the article is interesting and stimulating. Here you can find my comments:
OVERALL: - Please, enlarge and re-arrange font sizes to guide the reader properly in all sections. All figures must be composed of HD images. It is mandatory to improve the scientific quality of the whole manuscript.
- Please, pay attention to the JOURNAL TEMPLATE in all sections, including tables, references, captions, units, equations, and Figures.
- Please, improve the contrast between colours.
INTRODUCTION: Please, consider in the scientific background of your study the importance of evaluating models' accuracy in the management and monitoring of natural resources broadly speaking (i.e.,
Lama, G.F.C., Errico, A., Pasquino, V., Mirzaei, S., Preti, F., Chirico, G.B. 2022. Velocity uncertainty quantification based on Riparian vegetation indices in open channels colonized by Phragmites australis. J. Ecohydraulics 7(1), 71–76. https://doi.org/10.1080/24705357.2021.1938255.
Lama, G.F.C., Errico, A., Francalanci, S., Solari, L., Chirico, G.B., Preti, F. 2020. Hydraulic Modeling of Field Experiments in a Drainage Channel Under Different Riparian Vegetation Scenarios. In Innovative Biosystems Engineering for Sustainable Agriculture, Forestry and Food Production; Coppola A., Di Renzo G., Altieri G., D’Antonio P., Eds.; Springer: Cham, Switzerland, 2020; 69–77; doi:10.1007/978-3-030-39299-4_8.
Brunetti, G.F.A., Fallico, C., De Bartolo, S., Severino, G. (2022). Well‐Type Steady Flow in Strongly Heterogeneous Porous Media: An Experimental Study. Water Resour. Res., 58(5), e2021WR030717. doi.org/10.1029/2021WR030717.
Khan, M.A., Sharma, N., Lama, G.F.C., Hasan, M., Garg, R., Busico, G., Alharbi, R.S. 2022. Three-Dimensional Hole Size (3DHS) Approach for Water Flow Turbulence Analysis over Emerging Sand Bars: Flume-Scale Experiments. Water 14, 1889. https://doi.org/10.3390/w14121889)
METHODS: Please, insert a Figure for each sub-section.
Author Response
The topic of the article is interesting and stimulating. Here you can find my comments:
OVERALL: - Please, enlarge and re-arrange font sizes to guide the reader properly in all sections. All figures must be composed of HD images. It is mandatory to improve the scientific quality of the whole manuscript.
- Please, pay attention to the JOURNAL TEMPLATE in all sections, including tables, references, captions, units, equations, and Figures.
R/ Thank you very much for the comments, we tried to improve everything needed, so the quality of the manuscript fulfils the standards. The document was reorganized according to the JOURNAL TEMPLATE, including font type and size and sections. High quality images were included. Regarding the improvement of the scientific quality, we did our best to improve, but it seems to me that we need a little more time to improve everything.
- Please, improve the contrast between colours.
R/ The contrasts of the images were improved
INTRODUCTION: Please, consider in the scientific background of your study the importance of evaluating models' accuracy in the management and monitoring of natural resources broadly speaking (i.e.,
- Lama, G.F.C., Errico, A., Pasquino, V., Mirzaei, S., Preti, F., Chirico, G.B. 2022. Velocity uncertainty quantification based on Riparian vegetation indices in open channels colonized by Phragmites australis. J. Ecohydraulics 7(1), 71–76. https://doi.org/10.1080/24705357.2021.1938255.
- Lama, G.F.C., Errico, A., Francalanci, S., Solari, L., Chirico, G.B., Preti, F. 2020. Hydraulic Modeling of Field Experiments in a Drainage Channel Under Different Riparian Vegetation Scenarios. In Innovative Biosystems Engineering for Sustainable Agriculture, Forestry and Food Production; Coppola A., Di Renzo G., Altieri G., D’Antonio P., Eds.; Springer: Cham, Switzerland, 2020; 69–77; doi:10.1007/978-3-030-39299-4_8.
- Brunetti, G.F.A., Fallico, C., De Bartolo, S., Severino, G. (2022). Well‐Type Steady Flow in Strongly Heterogeneous Porous Media: An Experimental Study. Water Resour. Res., 58(5), e2021WR030717. doi.org/10.1029/2021WR030717.
- Khan, M.A., Sharma, N., Lama, G.F.C., Hasan, M., Garg, R., Busico, G., Alharbi, R.S. 2022. Three-Dimensional Hole Size (3DHS) Approach for Water Flow Turbulence Analysis over Emerging Sand Bars: Flume-Scale Experiments. Water 14, 1889. https://doi.org/10.3390/w14121889)
R/ Thank you very much for the literature recommendations, they were very important to understand the request. We are working in the suggestions as well as in the methodology and discussion comments about the need for calibration and validation. We also worked on the limitations in modelling and lacks in interpretations.
METHODS: Please, insert a Figure for each sub-section.
R/ We inserted the figures specifically the one related to the geographic scope. We moved the part of geographic scope to the introduction as suggested by one of the authors and worked in a new scheme with the flowchart.
Reviewer 2 Report
This is very interesting research about the impacts of nature-based solutions on the water availability for water supply. The authors use biophysical models to estimate the impact of six NBS implementation scenarios based on real data in the Cantareira Water Supply System (CWSS), which consists of 2,200 km2 of 52 source watersheds with four interconnected reservoirs with storage capacity of about 1,000 53 hm3, supplies water to nearly 9 million people in the Sao Paulo Metropolitan area.
The paper is well written; however, I have several comments that need to be addressed before publishing:
1. Abstract: the first sentence is misleading as there are no investments' estimation, only the impacts on water availability and water balance. Using acronyms and short names in the abstract, makes it difficult to read, so please avoid that. Also, you can add a sentence about the scenarios.. In short, the Abstract needs to be rewritten.
2. Introduction.
Ln 65,66: NbS are not equal to green infrastructure, so please only use NbS.
Ln 98 to 100: This paragraph needs revision. This is not the aim of the paper. But rather synthesizing a modelling framework for hydrological modelling to simulate impacts of NbS scenarios… Please clarify what exactly is the aim and the scope of the paper. It seems like RIOS and FIESTA were previously applied and here within this research the generated scenarios were implemented in HEC-HMS and SWAT – please clarify this in the introduction.
3. METHODS
Correct the title from Methodos to Methods
Ln 292: you have (i) and (ii), but (iii) is missing (deep aquifer)
Try to add the following:
· A table explaining the models’ input data would improve the reading.
· scheme of conceptual models od HEC HMS and SWAT by making their differences obvious, and explaining why using both would be beneficial.
· A table of most significant models parameters' values (initial values and calibrated values)
· Better explain the calibration and validation results. Why in the case of HEC the results for Paiva-Jug are the poorest, i.e. the RMSE is highest and in the case of SWAT the RMSE is lowest. The relationship between RMSE and Nash-Sutcliffe is strange in Table 2 – these two measures are not proportional. A low RMSE would signify high NSH (this is the case in Table1, but not in Table 2). Please check these numbers or explain the measures in the text (if you are using them in different form -is RMSE normalized?).
4. Results and discussion
Why scenarios PtF and FtP are not shown in Figure 4?
5. Typos: there are quite a few typos in the text, e.g.: use UN instead of ONU; Methods instead of Methodos… . Please review the text carefully.
Author Response
- Abstract: the first sentence is misleading as there are no investments' estimation, only the impacts on water availability and water balance. Using acronyms and short names in the abstract, makes it difficult to read, so please avoid that. Also, you can add a sentence about the scenarios. In short, the Abstract needs to be rewritten.
R/ Thank you very much for all the comments, they were very important to improve the quality of our article. Regarding the first sentence, we agreed, and my confusion was because this article is part of a larger study that has to do with economic valuation, so we decided to separate the topics and I did not update these sentences. We rewrote the abstract.
- Introduction.
Ln 65,66: NbS are not equal to green infrastructure, so please only use NbS.
R/ Done
Ln 98 to 100: This paragraph needs revision. This is not the aim of the paper. But rather synthesizing a modelling framework for hydrological modelling to simulate impacts of NbS scenarios… Please clarify what exactly is the aim and the scope of the paper.
R/ We modified the text
It seems like RIOS and FIESTA were previously applied and here within this research the generated scenarios were implemented in HEC-HMS and SWAT – please clarify this in the introduction.
R/ We incorporated a figure in the methodology to clarify this point and improved in the introduction with an explanation.
- METHODS
Correct the title from Methodos to Methods
R/ Done
Ln 292: you have (i) and (ii), but (iii) is missing (deep aquifer)
R/ Done
Try to add the following:
- A table explaining the models’ input data would improve the reading.
R/ I would need more time to create the table
- Scheme of conceptual models of HEC HMS and SWAT by making their differences obvious and explaining why using both would be beneficial.
R/ I would need more time to create the scheme of conceptual models
- A table of most significant model parameters' values (initial values and calibrated values)
R/ I’m working on it, but I need a little more time to incorporate in the document
- Better explain the calibration and validation results. Why in the case of HEC the results for Paiva-Jug are the poorest, i.e. the RMSE is highest and in the case of SWAT the RMSE is lowest. The relationship between RMSE and Nash-Sutcliffe is strange in Table 2 – these two measures are not proportional. A low RMSE would signify high NSH (this is the case in Table1, but not in Table 2). Please check these numbers or explain the measures in the text (if you are using them in different form -is RMSE normalized?).
R/ Thanks a lot for the observation, I changed the name of the statistics for SWAT and in the table it is not the RMSE but the R² that explains the differences. I will estimate the RMSE and change in the table and do some checking in the Hec-Hms with my colleague.
- Results and discussion
Why scenarios PtF and FtP are not shown in Figure 4?
R/ We added the scenarios in the Figure 4 (now Figure 6)
- Typos: there are quite a few typos in the text, e.g.: use UN instead of ONU; Methods instead of Methodos. Please review the text carefully.
R/ We need to finish the other recommendations to get a native English revisor to review the text carefully.
Round 2
Reviewer 1 Report
The article has been improved. Just few corrections are needed:
INTRODUCTION: Please, consider in the scientific background of your study the value of both advanced experimental and modeling analysis of the prediction of natural phenomena (i.e.,
Lama, G.F.C., Crimaldi, M. 2021. Assessing the role of Gap Fraction on the Leaf Area Index (LAI) estimations of riparian vegetation based on Fisheye lenses. 29th European Biomass Conference and Exhibition, 26-29 April 2021 Online, 1172–1176, 2021. https://dx.doi.org/10.5071/29thEUBCE2021-4AV.3.16.
Pirone, D., Cimorelli, L., Del Giudice, G., Pianese, D. 2022. Short-term rainfall forecasting using cumulative precipitation fields from station data: a probabilistic machine learning approach. J. Hydrol. 128949. https://doi.org/10.1016/j.jhydrol.2022.128949.
Crimaldi, M., Lama, G.F.C. 2021. Impacts of riparian plants biomass assessed by UAV-acquired multispectral images on the hydrodynamics of vegetated streams. 29th European Biomass Conference and Exhibition, 26-29 April 2021 Online, 1157–1161, 2021. https://dx.doi.org/10.5071/29thEUBCE2021-4AV.3.6)
Author Response
Thank you very much for the review. We add paragraphs in the introduction including scientific background on experimental analysis and on simulation models, as below:
Experimental studies with monitoring data carried out on the effects of NbS implementation in the watershed demonstrate that there is a relationship between land use land cover changes and the components of water flow. There is evidence of a decrease in peak discharge at the catchment outflow [6], an increase in water storage in the soil [7] and the improvement in water quality [8]. Data from remote sensing from 2000 to 2015 shows an increase in permanent surface water in vegetation restoration regions [9].
However, due to the complexity of the hydrological processes that occur in the basin it is valuable to use models to predict the long-term effects in the hydrological components. Some model evaluations indicate how reforestation projects, and LULC changes such us urbanization processes contribute significantly to hydrology [10]. In general, studies have shown that NbS can reduce surface runoff, favoring infiltration and decreasing soil erosion, contributing to establishing greater climatic resilience. [11].
We hope that this inclusion meets the request.
In addition, together with the coauthors we are still revising the English.
